# Decoupling Representation and Classifier for Long-Tailed Recognition

**Bingyi Kang[1,2], Saining Xie[1], Marcus Rohrbach[1], Zhicheng Yan[1], Albert Gordo[1], Jiashi Feng[2], Yannis Kalantidis[1]**
[1]Facebook AI, [2]National University of Singapore
kang@u.nus.edu,{s9xie,mrf,zyan3,agordo,yannisk}@fb.com,elefjia@nus.edu.sg

## Abstract

The long-tail distribution of the visual world poses great challenges for deep learning based classification models on how to handle the class imbalance problem. Existing solutions usually involve class-balancing strategies, *e.g.* by loss re-weighting, data re-sampling, or transfer learning from head- to tail-classes, but most of them adhere to the scheme of jointly learning representations and classifiers. In this work, we decouple the learning procedure into *representation learning* and *classification*, and systematically explore how different balancing strategies affect them for long-tailed recognition. The findings are surprising: (1) data imbalance might not be an issue in learning high-quality representations; (2) with representations learned with the simplest instance-balanced (natural) sampling, it is also possible to achieve strong long-tailed recognition ability by adjusting only the classifier. We conduct extensive experiments and set new state-of-the-art performance on common long-tailed benchmarks like ImageNet-LT, Places-LT and iNaturalist, showing that it is possible to outperform carefully designed losses, sampling strategies, even complex modules with memory, by using a straightforward approach that decouples representation and classification. Our code is available at https://github.com/facebookresearch/classifier-balancing.

## 1 Introduction

Visual recognition research has made rapid advances during the past years, driven primarily by the use of deep convolutional neural networks (CNNs) and large image datasets, most importantly the ImageNet Challenge (Russakovsky et al., 2015). Such datasets are usually artificially balanced with respect to the number of instances for each object/class in the training set. Visual phenomena, however, follow a long-tailed distribution that many standard approaches fail to properly model, leading to a significant drop in accuracy. Motivated by this, a number of works have recently emerged that try to study *long-tailed recognition*, *i.e.*, recognition in a setting where the number of instances in each class highly varies and follows a long-tailed distribution.

When learning with long-tailed data, a common challenge is that instance-rich (or head) classes dominate the training procedure. The learned classification model tends to perform better on these classes, while performance is significantly worse for instance-scarce (or tail) classes. To address this issue and to improve performance across all classes, one can re-sample the data or design specific loss functions that better facilitate learning with imbalanced data (Chawla et al., 2002; Cui et al., 2019; Cao et al., 2019). Another direction is to enhance recognition performance of the tail classes by transferring knowledge from the head classes (Wang et al., 2017; 2018; Zhong et al., 2019; Liu et al., 2019). Nevertheless, the common belief behind existing approaches is that designing proper sampling strategies, losses, or even more complex models, is useful for learning high-quality representations for long-tailed recognition.

Most aforementioned approaches thus learn the classifiers used for recognition jointly with the data representations. However, such a joint learning scheme makes it unclear how the long-tailed recognition ability is achieved—is it from learning a better representation or by handling the data imbalance better via shifting classifier decision boundaries? To answer this question, we take one step back and decouple long-tail recognition into *representation learning* and *classification*. For learning rep-

resentations, the model is exposed to the training instances and trained through different sampling strategies or losses. For classification, upon the learned representations, the model recognizes the long-tailed classes through various classifiers. We evaluate the performance of various sampling and classifier training strategies for long-tailed recognition under both joint and decoupled learning schemes.

Specifically, we first train models to learn representations with different sampling strategies, including the standard instance-based sampling, class-balanced sampling and a mixture of them. Next, we study three different basic approaches to obtain a classifier with balanced decision boundaries, on top of the learned representations. They are 1) re-training the parametric linear classifier in a class-balancing manner (*i.e.*, re-sampling); 2) non-parametric nearest class mean classifier, which classifies the data based on their closest class-specific mean representations from the training set; and 3) normalizing the classifier weights, which adjusts the weight magnitude directly to be more balanced, adding a temperature to modulate the normalization procedure.

We conduct extensive experiments to compare the aforementioned instantiations of the decoupled learning scheme with the conventional scheme that jointly trains the classifier and the representations. We also compare to recent, carefully designed and more complex models, including approaches using memory (*e.g.*, OLTR (Liu et al., 2019)) as well as more sophisticated losses (Cui et al., 2019). From our extensive study across three long-tail datasets, ImageNet-LT, Places-LT and iNaturalist, we make the following intriguing observations:

- We find that decoupling representation learning and classification has surprising results that challenge common beliefs for long-tailed recognition: instance-balanced sampling learns the best and most generalizable representations.

- It is advantageous in long-tailed recognition to re-adjust the decision boundaries specified by the jointly learned classifier during representation learning: Our experiments show that this can either be achieved by retraining the classifier with class-balanced sampling or by a simple, yet effective, classifier weight normalization which has only a single hyper-parameter controlling the "temperature" and which does not require additional training.

- By applying the decoupled learning scheme to standard networks (*e.g.*, ResNeXt), we achieve significantly higher accuracy than well established state-of-the-art methods (different sampling strategies, new loss designs and other complex modules) on multiple long-tailed recognition benchmark datasets, including ImageNet-LT, Places-LT, and iNaturalist.

## 2 RELATED WORK

Long-tailed recognition has attracted increasing attention due to the prevalence of imbalanced data in real-world applications (Wang et al., 2017; Zhou et al., 2017; Mahajan et al., 2018; Zhong et al., 2019; Gupta et al., 2019). Recent studies have mainly pursued the following three directions:

**Data distribution re-balancing.** Along this direction, researchers have proposed to re-sample the dataset to achieve a more balanced data distribution. These methods include over-sampling (Chawla et al., 2002; Han et al., 2005) for the minority classes (by adding copies of data), under-sampling (Drummond et al., 2003) for the majority classes (by removing data), and class-balanced sampling (Shen et al., 2016; Mahajan et al., 2018) based on the number of samples for each class.

**Class-balanced Losses.** Various methods are proposed to assign different losses to different training samples for each class. The loss can vary at class-level for matching a given data distribution and improving the generalization of tail classes (Cui et al., 2019; Khan et al., 2017; Cao et al., 2019; Khan et al., 2019; Huang et al., 2019). A more fine-grained control of the loss can also be achieved at sample level, *e.g.* with Focal loss (Lin et al., 2017), Meta-Weight-Net (Shu et al., 2019), re-weighted training (Ren et al., 2018), or based on Bayesian uncertainty (Khan et al., 2019). Recently, Hayat et al. (2019) proposed to balance the classification regions of head and tail classes using an affinity measure to enforce cluster centers of classes to be uniformly spaced and equidistant.

**Transfer learning from head- to tail classes.** Transfer-learning based methods address the issue of imbalanced training data by transferring features learned from head classes with abundant training instances to under-represented tail classes. Recent work includes transferring the intra-class

variance (Yin et al., 2019) and transferring semantic deep features (Liu et al., 2019). However it is usually a non-trivial task to design specific modules (*e.g.* external memory) for feature transfer.

A benchmark for low-shot recognition was proposed by Hariharan & Girshick (2017) and consists of a representation learning phase without access to the low-shot classes and a subsequent low-shot learning phase. In contrast, the setup for long-tail recognition assumes access to both head and tail classes and a more continuous decrease in in class labels. Recently, Liu et al. (2019) and Cao et al. (2019) adopt re-balancing schedules that learn representation and classifier jointly within a two-stage training scheme. OLTR (Liu et al., 2019) uses instance-balanced sampling to first learn representations that are fine-tuned in a second stage with class-balanced sampling together with a memory module. LDAM (Cao et al., 2019) introduces a label-distribution-aware margin loss that expands the decision boundaries of few-shot classes. In Section 5 we exhaustively compare to OLTR and LDAM, since they report state-of-the-art results for the ImageNet-LT, Places-LT and iNaturalist datasets. In our work, we argue for *decoupling* representation and classification. We demonstrate that in a long-tailed scenario, this separation allows straightforward approaches to achieve high recognition performance, without the need for designing sampling strategies, balance-aware losses or adding memory modules.

## 3   LEARNING REPRESENTATIONS FOR LONG-TAILED RECOGNITION

For long-tailed recognition, the training set follows a long-tailed distribution over the classes. As we have less data about infrequent classes during training, the models trained using imbalanced datasets tend to exhibit under-fitting on the few-shot classes. But in practice we are interested in obtaining the model capable of recognizing all classes well. Various re-sampling strategies (Chawla et al., 2002; Shen et al., 2016; Cao et al., 2019), loss reweighting and margin regularization over few-shot classes are thus proposed. However, it remains unclear how they achieve performance improvement, if any, for long-tailed recognition. Here we systematically investigate their effectiveness by disentangling representation learning from classifier learning, in order to identify what indeed matters for long-tailed recognition.

**Notation.**   We define the notation used through the paper. Let $X = \{x_i, y_i\}, i \in \{1, \ldots, n\}$ be a training set, where $y_i$ is the label for data point $x_i$. Let $n_j$ denote the number of training sample for class $j$, and let $n = \sum_{j=1}^{C} n_j$ be the total number of training samples. Without loss of generality, we assume that the classes are sorted by cardinality in decreasing order, *i.e.*, if $i < j$, then $n_i \geq n_j$. Additionally, since we are in a long-tail setting, $n_1 \gg n_C$. Finally, we denote with $f(x; \theta) = z$ the representation for $x$, where $f(x; \theta)$ is implemented by a deep CNN model with parameter $\theta$. The final class prediction $\tilde{y}$ is given by a classifier function $g$, such that $\tilde{y} = \arg\max g(z)$. For the common case, $g$ is a linear classifier, *i.e.*, $g(z) = \boldsymbol{W}^\top z + \boldsymbol{b}$, where $\boldsymbol{W}$ denotes the classifier weight matrix, and $\boldsymbol{b}$ is the bias. We present other instantiations of $g$ in Section 4.

**Sampling strategies.**   In this section we present a number of sampling strategies that aim at re-balancing the data distribution for representation and classifier learning. For most sampling strategies presented below, the probability $p_j$ of sampling a data point from class $j$ is given by:

$$p_j = \frac{n_j^q}{\sum_{i=1}^{C} n_i^q}, \tag{1}$$

where $q \in [0, 1]$ and $C$ is the number of training classes. Different sampling strategies arise for different values of $q$ and below we present strategies that correspond to $q = 1$, $q = 0$, and $q = 1/2$.

*Instance-balanced sampling.*   This is the most common way of sampling data, where each training example has equal probability of being selected. For instance-balanced sampling, the probability $p_j^{\text{IB}}$ is given by Equation 1 with $q = 1$, *i.e.*, a data point from class $j$ will be sampled proportionally to the cardinality $n_j$ of the class in the training set.

*Class-balanced sampling.*   For imbalanced datasets, instance-balanced sampling has been shown to be sub-optimal (Huang et al., 2016; Wang et al., 2017) as the model under-fits for few-shot classes leading to lower accuracy, especially for balanced test sets. Class-balanced sampling has been used to alleviate this discrepancy, as, in this case, each *class* has an equal probability of being selected. The probability $p_j^{\text{CB}}$ is given by Eq. (1) with $q = 0$, *i.e.*, $p_j^{\text{CB}} = 1/C$. One can see this as a two-

stage sampling strategy, where first a class is selected uniformly from the set of classes, and then an instance from that class is subsequently uniformly sampled.

*Square-root sampling.* A number of variants of the previous sampling strategies have been explored. A commonly used variant is square-root sampling (Mikolov et al., 2013; Mahajan et al., 2018), where $q$ is set to $1/2$ in Eq. (1) above.

*Progressively-balanced sampling.* Recent approaches (Cui et al., 2018; Cao et al., 2019) utilized mixed ways of sampling, *i.e.*, combinations of the sampling strategies presented above. In practice this involves first using instance-balanced sampling for a number of epochs, and then class-balanced sampling for the last epochs. These mixed sampling approaches require setting the number of epochs before switching the sampling strategy as an explicit hyper-parameter. Here, we experiment with a softer version, progressively-balanced sampling, that progressively "interpolates" between instance-balanced and class-balanced sampling as learning progresses. Its sampling probability/weight $p_j$ for class $j$ is now a function of the epoch $t$,

$$p_j^{\text{PB}}(t) = (1 - \frac{t}{T})p_j^{\text{IB}} + \frac{t}{T}p_j^{\text{CB}}, \qquad (2)$$

where $T$ is the total number of epochs. Figure 3 in appendix depicts the sampling probabilities.

**Loss re-weighting strategies.** Loss re-weighting functions for imbalanced data have been extensively studied, and it is beyond the scope of this paper to examine all related approaches. What is more, we found that some of the most recent approaches reporting high performance were hard to train and reproduce and in many cases require extensive, dataset-specific hyper-parameter tuning. In Section A of the Appendix we summarize the latest, best performing methods from this area. In Section 5 we show that, without bells and whistles, baseline methods equipped with a properly balanced classifier can perform equally well, if not better, than the latest loss re-weighting approaches.

## 4 CLASSIFICATION FOR LONG-TAILED RECOGNITION

When learning a classification model on balanced datasets, the classifier weights $W$ and $b$ are usually trained jointly with the model parameters $\theta$ for extracting the representation $f(x_i; \theta)$ by minimizing the cross-entropy loss between the ground truth $y_i$ and prediction $W^\top f(x_i; \theta) + b$. This is also a typical baseline for long-tailed recognition. Though various approaches of re-sampling, re-weighting and transferring representations from head to tail classes have been proposed, the general scheme remains the same: classifiers are either learned jointly with the representations either end-to-end, or via a two-stage approach where the classifier and the representation are jointly fine-tuned with variants of class-balanced sampling as a second stage (Cui et al., 2018; Cao et al., 2019).

In this section, we consider decoupling the representation from the classification in long-tailed recognition. We present ways of learning classifiers aiming at rectifying the decision boundaries on head- and tail-classes via fine-tuning with different sampling strategies or other non-parametric ways such as nearest class mean classifiers. We also consider an approach to rebalance the classifier weights that exhibits a high long-tailed recognition accuracy without any additional retraining.

**Classifier Re-training (cRT).** A straightforward approach is to re-train the classifier with class-balanced sampling. That is, keeping the representations fixed, we randomly re-initialize and optimize the classifier weights $W$ and $b$ for a small number of epochs using class-balanced sampling. A similar methodology was also recently used in (Zhang et al., 2019) for action recognition on a long-tail video dataset.

**Nearest Class Mean classifier (NCM).** Another commonly used approach is to first compute the mean feature representation for each class on the training set and then perform nearest neighbor search either using cosine similarity or the Euclidean distance computed on $L_2$ normalized mean features (Snell et al., 2017; Guerriero et al., 2018; Rebuffi et al., 2017). Despite its simplicity, this is a strong baseline (*cf*. the experimental evaluation in Section 5); the cosine similarity alleviates the weight imbalance problem via its inherent normalization (see also Figure 4).

**$\tau$-normalized classifier ($\tau$-normalized).** We investigate an efficient approach to re-balance the decision boundaries of classifiers, inspired by an empirical observation: after joint training with instance-balanced sampling, the norms of the weights $\|w_j\|$ are correlated with the cardinality of the

classes $n_j$, while, after fine-tuning the classifiers using class-balanced sampling, the norms of the classifier weights tend to be more similar (*cf*. Figure 2-left).

Inspired by the above observations, we consider rectifying imbalance of decision boundaries by adjusting the classifier weight norms directly through the following $\tau$-normalization procedure. Formally, let $\boldsymbol{W} = \{w_j\} \in \mathbb{R}^{d \times C}$, where $w_j \in \mathbb{R}^d$ are the classifier weights corresponding to class $j$. We scale the weights of $\boldsymbol{W}$ to get $\widetilde{\boldsymbol{W}} = \{\widetilde{w_j}\}$ by:

$$\widetilde{w_i} = \frac{w_i}{||w_i||^\tau}, \tag{3}$$

where $\tau$ is a hyper-parameter controlling the "temperature" of the normalization, and $||\cdot||$ denotes the $L_2$ norm. When $\tau = 1$, it reduces to standard $L_2$-normalization. When $\tau = 0$, no scaling is imposed. We empirically choose $\tau \in (0, 1)$ such that the weights can be rectified smoothly. After $\tau$-normalization, the classification logits are given by $\hat{y} = \widetilde{\boldsymbol{W}}^\top f(x; \theta)$. Note that we discard the bias term $\boldsymbol{b}$ here due to its negligible effect on the logits and final predictions.

**Learnable weight scaling (LWS).** Another way of interpreting $\tau$-normalization would be to think of it as a re-scaling of the magnitude for each classifier $w_i$ keeping the direction unchanged. This could be written as

$$\widetilde{w_i} = f_i * w_i, \text{where } f_i = \frac{1}{||w_i||^\tau}. \tag{4}$$

Although for $\tau$-normalized in general $\tau$ is chosen through cross-validation, we further investigate learning $f_i$ on the training set, using class-balanced sampling (like cRT). In this case, we keep both the representations and classifier weights fixed and only learn the scaling factors $f_i$. We denote this variant as *Learnable Weight Scaling* (LWS) in our experiments.

## 5 EXPERIMENTS

### 5.1 EXPERIMENTAL SETUP

**Datasets.** We perform extensive experiments on three large-scale long-tailed datasets, including Places-LT (Liu et al., 2019), ImageNet-LT (Liu et al., 2019), and iNaturalist 2018 (iNatrualist, 2018). Places-LT and ImageNet-LT are artificially truncated from their balanced versions (Places-2 (Zhou et al., 2017) and ImageNet-2012 (Deng et al., 2009)) so that the labels of the training set follow a long-tailed distribution. Places-LT contains images from 365 categories and the number of images per class ranges from 4980 to 5. ImageNet-LT has 1000 classes and the number of images per class ranges from 1280 to 5 images. iNaturalist 2018 is a real-world, naturally long-tailed dataset, consisting of samples from 8,142 species.

**Evaluation Protocol.** After training on the long-tailed datasets, we evaluate the models on the corresponding balanced test/validation datasets and report the commonly used top-1 accuracy over all classes, denoted as *All*. To better examine performance variations across classes with different number of examples seen during training, we follow Liu et al. (2019) and further report accuracy on three splits of the set of classes: *Many-shot* (more than 100 images), *Medium-shot* (20∼100 images) and *Few-shot* (less than 20 images). Accuracy is reported as a percentage.

**Implementation.** We use the PyTorch (Paszke et al., 2017) framework for all experiments[1]. For Places-LT, we choose ResNet-152 as the backbone network and pretrain it on the full ImageNet-2012 dataset, following Liu et al. (2019). On ImageNet-LT, we report results with ResNet-{10,50,101,152} (He et al., 2016) and ResNeXt-{50,101,152}(32x4d) (Xie et al., 2017) but mainly use ResNeXt-50 for analysis. Similarly, ResNet-{50,101,152} is also used for iNaturalist 2018. For all experiements, if not specified, we use SGD optimizer with momentum 0.9, batch size 512, cosine learning rate schedule (Loshchilov & Hutter, 2016) gradually decaying from 0.2 to 0 and image resolution $224 \times 224$. In the first representation learning stage, the backbone network is usually trained for 90 epochs. In the second stage, *i.e.*, for retraining a classifier (cRT), we restart the learning rate and train it for 10 epochs while keeping the backbone network fixed.

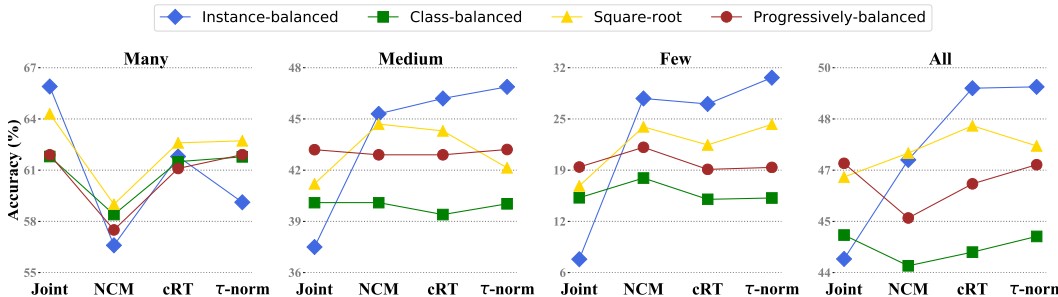

Figure 1: The performance of different classifiers for each split on ImageNet-LT with ResNeXt-50. Colored markers denote the sampling strategies used to learn the representations.

## 5.2 SAMPLING STRATEGIES AND DECOUPLED LEARNING

In Figure 1, we compare different sampling strategies for the conventional joint training scheme to a number of variations of the decoupled learning scheme on the ImageNet-LT dataset. For the joint training scheme (Joint), the linear classifier and backbone for representation learning are jointly trained for 90 epochs using a standard cross-entropy loss and different sampling strategies, *i.e.*, Instance-balanced, Class-balanced, Square-root, and Progressively-balanced. For the decoupled learning schemes, we present results when learning the classifier in all the ways presented in Section 4, *i.e.*, re-initialize and re-train (cRT), Nearest Class Mean (NCM) as well as $\tau$-normalized classifier. Below, we discuss a number of key observations.

*Sampling matters when training jointly.* From the Joint results in Figure 1 across sampling methods and splits, we see consistent gains in performance when using better sampling strategies (see also Table 5). The trends are consistent for the overall performance as well as the medium- and few-shot classes, with progressively-balanced sampling giving the best results. As expected, instance-balanced sampling gives the highest performance for the many-shot classes. This is well expected since the resulted model is highly skewed to the many-shot classes. Our results for different sampling strategies on joint training validate related works that try to design better data sampling methods.

*Joint or decoupled learning?* For most cases presented in Figure 1, performance using decoupled methods is significantly better in terms of overall performance, as well as all splits apart from the many-shot case. Even the non-parametric NCM approach is highly competitive in most cases, while cRT and $\tau$-normalized outperform the jointly trained baseline by a large margin (*i.e.* 5% higher than the jointly learned classifier), and even achieving 2% higher overall accuracy than the best jointly trained setup with progressively-balanced sampling. The gains are even higher for medium- and few-shot classes at 5% and 11%, respectively.

Table 1: Retraining/finetuning different parts of a ResNeXt-50 model on ImageNet-LT. B: backbone; C: classifier; LB: last block.

| Re-train | Many | Medium | Few | All |
|---|---|---|---|---|
| B+C | 55.4 | 45.3 | 24.5 | 46.3 |
| B+C(0.1×lr) | **61.9** | 45.6 | 22.8 | 48.8 |
| LB+C | 61.4 | 45.8 | 24.5 | 48.9 |
| C | 61.5 | **46.2** | **27.0** | **49.5** |

To further justify our claim that it is beneficial to decouple representation and classifier, we experiment with fine-tuning the backbone network (ResNeXt-50) jointly with the linear classifier. In Table 1, we present results when fine-tuning the whole network with standard or smaller (0.1×) learning rate, fine-tuning only the last block in the backbone, or only retraining the linear classifier and fixing the representation. Fine-tuning the whole network yields the worst performance (46.3% and 48.8%), while keeping the representation frozen performs best (49.5%). The trend is even more evident for the medium/few-shot classes. This result suggests that decoupling representation and classifier is desirable for long-tailed recognition.

---

[1]We will open-source our codebase and models.

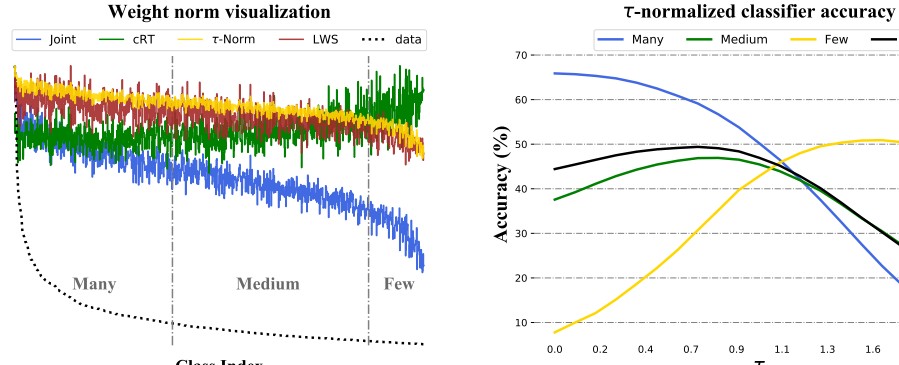

Figure 2: Left: Classifier weight norms for ImageNet-LT validation set when classes are sorted by descending values of $n_j$. Blue line: classifier weights learned with instance-balanced sampling. Green line: weights after fine-tuning with class-balanced sampling. Gold line: after $\tau$ normalization. Brown line: weights by learnable weight scaling. Right: Accuracy with different values of the normalization parameter $\tau$.

*Instance-balanced sampling gives the most generalizable representations.* Among all decoupled methods, when it comes to overall performance and all splits apart from the many-shot classes, we see that Instance-balanced sampling gives the best results. This is particularly interesting, as it implies that *data imbalance might not be an issue learning high-quality representations*.

## 5.3 How to balance your classifier?

Among the ways of balancing the classifier explored in Figure 1, the non-parametric NCM seems to perform slightly worse than cRT and $\tau$-normalization. Those two methods are consistently better in most cases apart from the few-shot case, where NCM performs comparably. The biggest drop for the NCM approach comes from the many-shot case. It is yet still somehow surprising that both the NCM and $\tau$-normalized cases give competitive performance even though they are free of additional training and involve no additional sampling procedure. As discussed in Section 4, their strong performance may stem from their ability to adaptively adjust the decision boundaries for many-, medium- and few-shot classes (see also Figure 4).

In Figure 2 (left) we empirically show the $L_2$ norms of the weight vectors for all classifiers, as well as the training data distribution sorted in a descending manner with respect to the number of instances in the training set. We can observe that the weight norm of the joint classifier (blue line) is positively correlated with the number of training instances of the corresponding class. More-shot classes tend to learn a classifier with larger magnitudes. As illustrated in Figure 4, this yields a wider classification boundary in feature space, allowing the classifier to have much higher accuracy on data-rich classes, but hurting data-scarce classes. $\tau$-normalized classifiers (gold line) alleviate this issue to some extent by providing more balanced classifier weight magnitudes. For retraining (green line), the weights are almost balanced except that few-shot classes have slightly larger classifier weight norms. Note that the NCM approach would give a horizontal line in the figure as the mean vectors are $L_2$-normalized before nearest neighbor search.

In Figure 2 (right), we further investigate how the performance changes as the temperature parameter $\tau$ for the $\tau$-normalized classifier varies. The figure shows that as $\tau$ increases from 0, many-shot accuracy decays dramatically while few-shot accuracy increases dramatically.

## 5.4 Comparison with the state-of-the-art on long-tailed datasets

In this section, we compare the performance of the decoupled schemes to other recent works that report state-of-the-art results on on three common long-tailed benchmarks: ImageNet-LT, iNaturalist and Places-LT. Results are presented in Tables 2, 3 and 4, respectively.

Table 2: Long-tail recognition accuracy on ImageNet-LT for different backbone architectures. † denotes results directly copied from Liu et al. (2019). * denotes results reproduced with the authors' code. ** denotes OLTR with our representation learning stage.

| Method | ResNet-10 | ResNeXt-50 | ResNeXt-152 |
|---|---|---|---|
| FSLwF† (Gidaris & Komodakis, 2018) | 28.4 | - | - |
| Focal Loss† (Lin et al., 2017) | 30.5 | - | - |
| Range Loss† (Zhang et al., 2017) | 30.7 | - | - |
| Lifted Loss† (Oh Song et al., 2016) | 30.8 | - | - |
| OLTR† (Liu et al., 2019) | 35.6 | - | - |
| OLTR* | 34.1 | 37.7 | 24.8 |
| OLTR** | 37.3 | 46.3 | 50.3 |
| Joint | 34.8 | 44.4 | 47.8 |
| NCM | 35.5 | 47.3 | 51.3 |
| cRT | **41.8** | **49.5** | 52.4 |
| $\tau$-normalized | 40.6 | **49.4** | **52.8** |
| LWS | **41.4** | **49.9** | **53.3** |

**ImageNet-LT.** Table 2 presents results for ImageNet-LT. Although related works present results with ResNet-10 (Liu et al., 2019), we found that using bigger backbone architectures increases performance significantly on this dataset. We therefore present results for three backbones: ResNet-10, ResNeXt-50 and the larger ResNeXt-152. For the state-of-the-art OLTR method of Liu et al. (2019) we adopt results reported in the paper, as well as results we reproduced using the authors' open-sourced codebase[2] with two training settings: the one suggested in the codebase and the one using our training setting for the representation learning. From the table we see that the non-parametric decoupled NCM method performs on par with the state-of-the-art for most architectures. We also see that when re-balancing the classifier properly, either by re-training or $\tau$-normalizing, we get results that, without bells and whistles outperform the current state-of-the-art for all backbone architectures. We further experimented with adding the memory mechanism of Liu et al. (2019) on top of our decoupled cRT setup, but the memory mechanism didn't seem to further boost performance (see Appendix B.4).

**iNaturalist 2018.** We further evaluate our decoupled methods on the iNaturalist 2018 dataset. We present results after 90 and 200 epochs, as we found that 90 epochs were not enough for the representation learning stage to converge; this is different from Cao et al. (2019) where they train for 90 epochs. From Table 3 we see that results are consistent with the ImageNet-LT case: re-balancing the classifier gives results that outperform CB-Focal (Cui et al., 2019). Our performance, when training only for 90 epochs, is slightly lower than the very recently proposed LDAM+DRW (Cao et al., 2019). However, with 200 training epochs and classifier normalization, we achieve a new state-of-the-art of 69.3 with ResNet-50 that can be further improved to 72.5 for ResNet-152. It is further worth noting that we cannot reproduce the numbers reported in Cao et al. (2019). We find that the $\tau$-normalized classifier performs best and gives a new state-of-the-art for the dataset, while surprisingly achieving similar accuracy (69%/72% for ResNet-50/ResNet-152) *across all many-, medium- and few-shot class splits*, a highly desired result for long-tailed recognition. Complete results, *i.e.*, for all splits and more backbone architectures can be found in Table 8 of the Appendix.

**Places-LT.** For Places-LT we follow the protocol of Liu et al. (2019) and start from a ResNet-152 backbone pre-trained on the *full* ImageNet dataset. Similar to Liu et al. (2019), we then fine-tune the backbone with Instance-balanced sampling for representation learning. Classification follows with fixed representations for our decoupled methods. As we see in Table 4, all three decoupled methods outperform the state-of-the-art approaches, including Lifted Loss (Oh Song et al., 2016), Focal Loss (Lin et al., 2017), Range Loss (Zhang et al., 2017), FSLwF (Gidaris & Komodakis, 2018) and OLTR (Liu et al., 2019). Once again, the $\tau$-normalized classifier give the top performance, with impressive gains for the medium- and few-shot classes.

---

[2] https://github.com/zhmiao/OpenLongTailRecognition-OLTR

Table 3: Overall accuracy on iNaturalist 2018. Rows with † denote results directly copied from Cao et al. (2019). We present results when training for 90/200 epochs.

| Method | ResNet-50 | ResNet-152 |
|---|---|---|
| CB-Focal† | 61.1 | - |
| LDAM† | 64.6 | - |
| LDAM+DRW† | 68.0 | - |
| Joint | 61.7/65.8 | 65.0/69.0 |
| NCM | 58.2/63.1 | 61.9/67.3 |
| cRT | 65.2/67.6 | 68.5/71.2 |
| $\tau$-normalized | 65.6/**69.3** | 68.8/**72.5** |
| LWS | 65.9/**69.5** | 69.1/72.1 |

Table 4: Results on Places-LT, starting from an ImageNet pre-trained ResNet152. † denotes results directly copied from Liu et al. (2019).

| Method | Many | Medium | Few | All |
|---|---|---|---|---|
| Lifted Loss† | 41.1 | 35.4 | 24.0 | 35.2 |
| Focal Loss† | 41.1 | 34.8 | 22.4 | 34.6 |
| Range Loss† | 41.1 | 35.4 | 23.2 | 35.1 |
| FSLwF† | 43.9 | 29.9 | 29.5 | 34.9 |
| OLTR† | 44.7 | 37.0 | 25.3 | 35.9 |
| Joint | **45.7** | 27.3 | 8.2 | 30.2 |
| NCM | 40.4 | 37.1 | 27.3 | 36.4 |
| cRT | 42.0 | 37.6 | 24.9 | 36.7 |
| $\tau$-normalized | 37.8 | **40.7** | **31.8** | **37.9** |
| LWS | 40.6 | 39.1 | 28.6 | 37.6 |

## 6 CONCLUSIONS

In this work, we explore a number of learning schemes for long-tailed recognition and compare jointly learning the representation and classifier to a number of straightforward decoupled methods. Through an extensive study we find that although sampling strategies matter when jointly learning representation and classifiers, instance-balanced sampling gives more generalizable representations that can achieve state-of-the-art performance after properly re-balancing the classifiers and without need of carefully designed losses or memory units. We set new state-of-the-art performance for three long-tailed benchmarks and believe that our findings not only contribute to a deeper understanding of the long-tailed recognition task, but can offer inspiration for future work.

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

## A    LOSS RE-WEIGHTING STRATEGIES

Here, we summarize some of the best performing loss re-weighting methods that we compare against in Section 5. Introduced in the context of object detection where imbalance exists in most common benchmarks, the Focal loss (Lin et al., 2017) aims to balance the sample-wise classification loss for model training by down-weighing easy samples. To this end, given a probability prediction $h_i$ for the sample $x_i$ over its true category $y_i$, it adds a re-weighting factor $(1 - h_i)^\gamma$ with $\gamma > 0$ into the standard cross-entropy loss $\mathcal{L}_{CE}$:

$$\mathcal{L}_{\text{focal}} := (1 - h_i)^\gamma \mathcal{L}_{CE} = -(1 - h_i)^\gamma \log(h_i). \tag{5}$$

For easy samples (which may dominate the training samples) with large predicted probability $h_i$ for their true categories, their corresponding cross entropy loss will be down weighted. Recently, Cui et al. (2019) presented a class balanced variant of the focal loss and applied it to long-tailed recognition. They modulated the Focal loss for a sample from class $j$ with a balance-aware coefficient equal to $(1 - \beta)/(1 - \beta_j^n)$. Very recently, Cao et al. (2019) proposed a label-distribution-aware margin (LDAM) loss that encourages few-shot classes to have larger margins, and their final loss is formulated as a cross-entropy loss with enforced margins:

$$\mathcal{L}_{\text{LDAM}} := -\log \frac{e^{\hat{y}_j - \Delta_j}}{e^{\hat{y}_j - \Delta_j} + \sum_c \neq j e^{\hat{y}_c - \Delta_c}}, \tag{6}$$

where $\hat{y}$ are the logits and $\Delta_j$ is a class-aware margin, inversely proportional to $n_j^{1/4}$.

## B    FURTHER ANALYSIS AND RESULTS

### B.1    SAMPLING STRATEGIES

In Figure 3 we visualize the sampling weights for the four sampling strategies we explore. In Table 5 we present accuracy on ImageNet-LT for "all" classes when training the representation and classifier jointly. It is clear that better sampling strategies help when jointly training the classifier with the representations/backbone architecture.

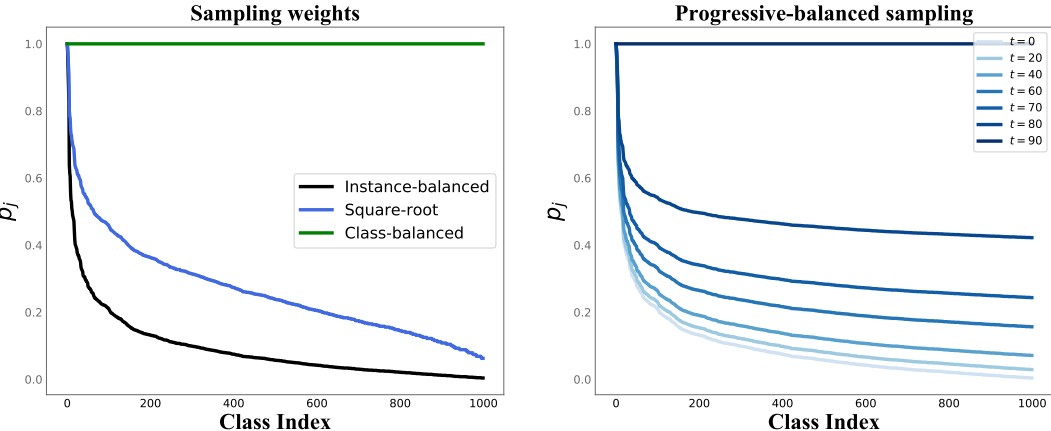

Figure 3: Sampling weights $p_j$ for ImageNet-LT. Classes are ordered with decreasing $n_j$ on the x-axis. Left: instance-balanced, class-balanced and square-root sampling. Right: Progressively-balanced sampling; as epochs progress, sampling goes from instance-balanced to class-balanced sampling.

### B.2    CLASSIFIER DECISION BOUNDARIES FOR $\tau$-NORMALIZED AND NCM

In Figure 4 we illustrate the classifier decision boundaries before/after normalization with Eq.(3), as well as when using cosine distance. Balancing the norms also leads to more balanced decision boundaries, allowing the classifiers for few-shot classes to occupy more space.

Table 5: Accuracy on ImageNet-LT when jointly learning the representation and classifier using different sampling strategies. Results in this Table are a subset of the results presented in Figure 1.

| Sampling | Many | Medium | Few | All |
|---|---|---|---|---|
| Instance-balanced | **65.9** | 37.5 | 7.7 | 44.4 |
| Class-balanced | 61.8 | 40.1 | 15.5 | 45.1 |
| Square-root | 64.3 | 41.2 | 17.0 | 46.8 |
| Progressively-balanced | 61.9 | **43.2** | **19.4** | **47.2** |

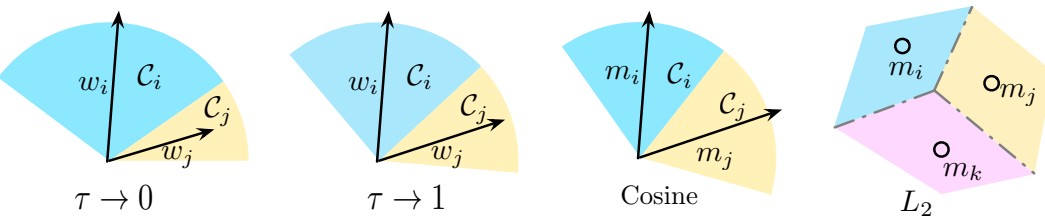

Figure 4: Illustrations on different classifiers and their corresponding decision boundaries, where $w_i$ and $w_j$ denote the classification weight for class $i$ and $j$ respectively, $\mathcal{C}_i$ is the classification cone belongs to class $i$ in the feature space, $m_i$ is the feature mean for class $i$. From left to right: $\tau$-normalized classifiers with $\tau \to 0$: the classifier with larger weights have wider decision boundaries; $\tau$-normalized classifiers with $\tau \to 1$: the decision boundaries are more balanced for different classes; NCM with cosine-similarity whose decision boundary is independent of the classifier weights; NCM with Euclidean-similarity whose decision boundaries partition the feature space into Voronoi cells.

## B.3 CLASSIFIER LEARNING COMPARISON TABLE

Table 6 presents some comparative analysis for the four different ways of learning the classifier that are presented in Section 4.

## B.4 VARYING THE BACKBONE ARCHITECTURE SIZE

**ImageNet-LT.** In Figure 5 we compare the performance of different backbone architecture sizes (model capacity) under different methods, including of different methods 1) OLTR (Liu et al., 2019) using the authors' codebase settings (OLTR*); 2) OLTR using the representation learning stage detailed in Section 5 (OLTR**); 3) cRT with the memory module from Liu et al. (2019) while training the classifier; 4) cRT; and 5) $\tau$-normalized. we see that a) the authors' implementation of OLTR over-fits for larger models, b) overfitting can be alleviated with our training setup (different training and LR schedules) c) adding the memory unit when re-training the classifier doesn't increase performance. Additional results of Table 2 are given in Table 7.

**iNaturalist 2018.** In Table 8 we present an extended version of the results of Table 3. We show results per split as well as results with a ResNet-101 backbone. As we see from the table and mentioned in Section 5, training only for 90 epochs gives sub-optimal representations, while both large models and longer training result in much higher accuracy on this challenging, large-scale task. What is even more interesting, we see performance across the many-, medium- and few-shot splits being approximately equal after re-balancing the classifier, with only a small advantage for the many-shot classes.

| | Joint | NCM | cRT | $\tau$-normalized | LWS |
|---|---|---|---|---|---|
| Decoupled from repr. | ✗ | ✓ | ✓ | ✓ | ✓ |
| No extra training | ✓ | ✓ | ✗ | ✓ | ✗ |
| No extra hyper-parameters | ✓ | ✓ | ✓ | ✗ | ✓ |
| Performance | ★ | ★★ | ★★★ | ★★★ | ★★★ |

Table 6: Comparative analysis for different ways of learning the classifier for long-tail recognition.

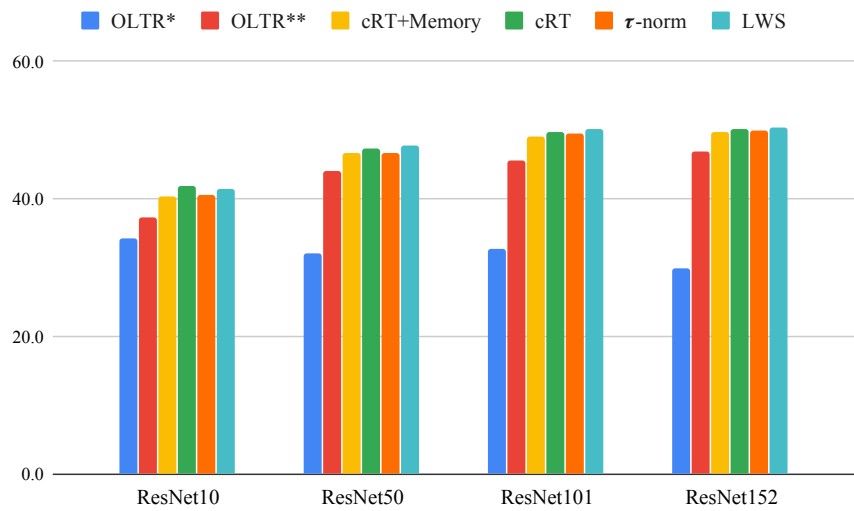

Figure 5: Accuracy on ImageNet-LT for different backbones

Table 7: Comprehensive results on ImageNet-LT with different backbone networks {ResNet, ResNeXt}-{50, 101,152}

| Backbone | Method | ResNet | | | | ResNeXt | | | |
|---|---|---|---|---|---|---|---|---|---|
| | | Many | Medium | Few | All | Many | Medium | Few | All |
| *-50 | Joint | 64.0 | 33.8 | 5.8 | 41.6 | 65.9 | 37.5 | 7.7 | 44.4 |
| | NCM | 53.1 | 42.3 | 26.5 | 44.3 | 56.6 | 45.3 | 28.1 | 47.3 |
| | cRT | 58.8 | 44.0 | 26.1 | 47.3 | 61.8 | 46.2 | 27.4 | 49.6 |
| | $\tau$-normalized | 56.6 | 44.2 | 27.4 | 46.7 | 59.1 | 46.9 | 30.7 | 49.4 |
| | LWS | 57.1 | 45.2 | 29.3 | 47.7 | 60.2 | 47.2 | 30.3 | 49.9 |
| *-101 | Joint | 66.6 | 36.8 | 7.1 | 44.2 | 66.2 | 37.8 | 8.6 | 44.8 |
| | NCM | 56.8 | 45.1 | 28.8 | 47.4 | 57.2 | 45.5 | 29.5 | 47.8 |
| | cRT | 61.6 | 46.5 | 28.0 | 49.8 | 61.7 | 46.0 | 27.0 | 49.4 |
| | $\tau$-normalized | 59.4 | 47.0 | 30.6 | 49.6 | 59.1 | 47.0 | 31.7 | 49.6 |
| | LWS | 60.1 | 47.6 | 31.2 | 50.2 | 60.5 | 47.2 | 31.2 | 50.1 |
| *-152 | Joint | 66.9 | 27.7 | 7.7 | 44.9 | 69.1 | 41.4 | 10.4 | 47.8 |
| | NCM | 56.9 | 45.6 | 29.9 | 47.8 | 60.3 | 49.0 | 33.6 | 51.3 |
| | cRT | 61.8 | 46.8 | 28.4 | 50.1 | 64.7 | 49.1 | 29.4 | 52.4 |
| | $\tau$-normalized | 59.6 | 47.5 | 32.2 | 50.1 | 62.2 | 50.1 | 35.8 | 52.8 |
| | LWS | 60.6 | 47.8 | 31.4 | 50.5 | 63.5 | 50.4 | 34.2 | 53.3 |

## B.5 ON THE EXPLORATION OF DETERMINING $\tau$

The current tau-normalization strategy does require a validation set to choose tau, which could be a disadvantage depending on the practical scenario. Can we do better?

**Finding $\tau$ value on training set.** We also attempted to select $\tau$ directly on the *training* dataset. Surprisingly, final performance on testing set is very similar, with $\tau$ selected using training set only.

We achieve this goal by simulating a balanced testing distribution from the training set. We first feed the whole training set through the network to get the top-1 accuracy for each of the classes. Then, we average the class-specific accuracies and use the averaged accuracy as the metric to determine the tau value. As shown in Table 9, we compare the $\tau$ found on training set and validation set for all three datasets. We can see that both the vale of $\tau$ and the overall performances are very close to each other, which demonstrates the effectiveness of searching for $\tau$ on training set. This strategy offers a practical way to find $\tau$ even when validation set is not available.

**Learning $\tau$ value on training set.** We further investigate if we can automatically learn the $\tau$ value instead of grid search. To this end, following cRT, we set $\tau$ as a learnable parameter and learn it on the training set with balanced sampling, while keeping all the other parameters fixed (including both the backbone network and classifier). Also, we compare the learned $\tau$ value and the corresponding results in the Table 9 (denoted by "learn" = ✓). This further reduces the manual effort of searching best $\tau$ values and make the strategy more accessible for practical usage.

Table 8: Comprehensive results on iNaturalist 2018 with different backbone networks (ResNet-50, ResNet-101 & ResNet-152) and different training epochs (90 & 200)

| Backbone | Method | 90 Epochs | | | | 200 Epochs | | | |
|---|---|---|---|---|---|---|---|---|---|
| | | Many | Medium | Few | All | Many | Medium | Few | All |
| ResNet-50 | Joint | 72.2 | 63.0 | 57.2 | 61.7 | 75.7 | 66.9 | 61.7 | 65.8 |
| | NCM | 55.5 | 57.9 | 59.3 | 58.2 | 61.0 | 63.5 | 63.3 | 63.1 |
| | cRT | 69.0 | 66.0 | 63.2 | 65.2 | 73.2 | 68.8 | 66.1 | 68.2 |
| | $\tau$-normalized | 65.6 | 65.3 | 65.9 | 65.6 | 71.1 | 68.9 | 69.3 | 69.3 |
| | LWS | 65.0 | 66.3 | 65.5 | 65.9 | 71.0 | 69.8 | 68.8 | 69.5 |
| ResNet-101 | Joint | 75.9 | 66.0 | 59.9 | 64.6 | 75.5 | 68.9 | 63.2 | 67.3 |
| | NCM | 58.6 | 61.9 | 61.8 | 61.5 | 63.7 | 65.7 | 65.3 | 65.3 |
| | cRT | 73.0 | 68.9 | 65.7 | 68.1 | 73.9 | 70.4 | 67.8 | 69.7 |
| | $\tau$-normalized | 69.7 | 68.3 | 68.3 | 68.5 | 68.6 | 70.6 | 72.2 | 71.0 |
| | LWS | 69.6 | 69.1 | 67.9 | 68.7 | 71.5 | 71.3 | 69.7 | 70.7 |
| ResNet-152 | Joint | 75.2 | 66.3 | 60.7 | 65.0 | 78.2 | 70.6 | 64.7 | 69.0 |
| | NCM | 59.3 | 61.9 | 62.6 | 61.9 | 66.3 | 67.5 | 67.2 | 67.3 |
| | cRT | 73.6 | 69.3 | 66.3 | 68.5 | 75.9 | 71.9 | 69.1 | 71.2 |
| | $\tau$-normalized | 69.8 | 68.5 | 68.9 | 68.8 | 74.3 | 72.3 | 72.2 | 72.5 |
| | LWS | 69.4 | 69.5 | 68.6 | 69.1 | 74.3 | 72.4 | 71.2 | 72.1 |

Table 9: Determining $\tau$ on the training set

| Dataset | split | learn | $\tau$ | Many | Medium | Few | All |
|---|---|---|---|---|---|---|---|
| ImageNet-LT | val | ✗ | 0.7 | 59.1 | 46.9 | 30.7 | 49.4 |
| | train | ✗ | 0.7 | 59.1 | 46.9 | 30.7 | 49.4 |
| | train | ✓ | 0.6968 | 59.2 | 46.9 | 30.6 | 49.4 |
| iNaturalist | val | ✗ | 0.3 | 65.6 | 65.3 | 65.9 | 65.6 |
| | train | ✗ | 0.2 | 69.0 | 65.2 | 63.6 | 65.0 |
| | train | ✓ | 0.3146 | 65.1 | 65.2 | 66.1 | 65.6 |
| Places-LT | val | ✗ | 0.8 | 37.8 | 40.7 | 31.8 | 37.9 |
| | train | ✗ | 0.6 | 41.4 | 39.3 | 25.3 | 37.4 |
| | train | ✓ | 0.5246 | 42.6 | 38.3 | 22.7 | 36.8 |

## B.6 COMPARING MLP CLASSFIIER WITH LINEAR CLASSIFIER

We experimented with MLPs with different layers (2 or 3) and different number of hidden neurons (2048 or 512). We use ReLU as activation function, set the batch size to be 512, and train the MLP using balanced sampling on fixed representation for 10 epochs with a cosine learning rate schedule, which gradually decrease the learning rate to zero. We conducted experiments on two datasets.

On ImageNet-LT, we use ResNeXt50 as the backbone network. The results are summarized in Table 10. We can see that when the MLP going deeper, the performance are getting worse. It probably means the backbone network is enough to learn discriminative representation.

Table 10: MLP classifier on ImageNet-LT

| Layers | hid-dim : 2048 | | | | hid-dim : 512 | | | |
|---|---|---|---|---|---|---|---|---|
| | Many | Medium | Few | All | Many | Medium | Few | All |
| 1 | 61.7 | 45.9 | 26.8 | 49.4 | | | | |
| 2 | 60.8 | 44.4 | 24.5 | 48.0 | 59.9 | 44.3 | 25.1 | 47.7 |
| 3 | 60.3 | 44.3 | 23.7 | 47.7 | 59.3 | 43.7 | 23.9 | 47.0 |

For iNaturalist, we use the representation from a ResNet50 model trained for 200 epochs. We only consider a hidden dimension of 2048, as this dataset contains much more classes. The results are shown in Table 11, and show that performance drop is even more severe when a deeper classifier is used.

Table 11: MLP classifier on iNaturalst

| Layers | Many | Medium | Few | All |
|---|---|---|---|---|
| 1 | 73.2 | 68.8 | 66.1 | 68.2 |
| 2 | 60.4 | 61.8 | 60.6 | 61.2 |
| 3 | 68.5 | 63.6 | 60.1 | 62.8 |

## B.7 COSINE SIMILARITY FOR CLASSIFICATION

We tried to replace the linear classifier with a cosine similarity classifier with (denoted by "cos") and without (denoted by "cos(noRelu)") the last ReLU activation function, following Gidaris & Komodakis (2018). We summarize the results in Table 12, which show that they are comparable to each other.

Table 12: Cosine similarity Classifier

| Classifier | Many | Medium | Few | All |
|---|---|---|---|---|
| NCM | 56.6 | 45.3 | 28.1 | 47.3 |
| cRT | 61.7 | 45.9 | 26.8 | 49.4 |
| $\tau$-normalized | 59.1 | 46.9 | 30.7 | 49.4 |
| cos | 60.4 | 46.8 | 29.3 | 49.7 |
| cos(noRelu) | 60.7 | 46.9 | 28.0 | 49.6 |

