# OpenReview forum: "Decoupling Representation and Classifier for Long-Tailed Recognition"
_ICLR.cc/2020/Conference — Accept (Poster)_

### Official Review · AnonReviewer3 · 2019-10-20
**Official Blind Review #3**

**Rating:** 6

**Review:**

This paper proposes to tackle long-tailed classification by treating separately the representation learning and the creation of a classifier for test time. They evaluate their method on several standard long-tailed classification datasets like ImageNet-LT or Places-LT.

Pros:
* Very well presented and clear
* Thorough experiments with baselines and comparisons with competitors
* Novel and efficient approach of redesigning the classifier as a post-processing step after the representation training

Cons:
* I did not find any single value of the "temperature" coefficient that you use for the different datasets! According to Fig 2, it should be around 0.7 for ImageNet-LT but you should clearly specify the used values for all the datasets. For reproducibility. It is also important to know it as it has an impact on how useful is this approach in practice. Because if the coefficients are very different for all the datasets, then the method requires a validation set to find this hyperparameter.
* As middle point between NCM and cRT, you could also train a cosine classifier as done in the paper that you cite "Dynamic few-shot visual learning without forgetting" by Gidaris et al. There is pytorch code for it online.


I am leaning towards acceptance as the method is clear, easy to implement, well studied through the experiments and has good results on standard benchmarks. The paper also provides interesting insights about long-tailed recognition in general like the effect of the different samplings with the proposed method.

**Experience Assessment:**

I have read many papers in this area.

**Review Assessment: Checking Correctness Of Derivations And Theory:**

N/A

**Review Assessment: Checking Correctness Of Experiments:**

I carefully checked the experiments.

**Review Assessment: Thoroughness In Paper Reading:**

N/A

---

> ### Author Response · Authors · 2019-11-15
> **Response to Blind Review #3**
>
> A1: [tau normalization] Please refer to general response.
>
> A2: [train a cosine classifier as done in the paper that you cite "Dynamic few-shot visual learning without forgetting" by Gidaris et al.]
>
> We tried to replace the linear classifier with a cosine similarity classifier with (denoted by Cos in the table) and without (denoted by noRelu in the table) the last ReLU activation function. We can see the results are comparable to each other. Thank you for your suggestion, and we will add this to the revision.
> -------------------------------------------------------
> Classifier       Many  Medium   Few     All
> -------------------------------------------------------
>  NCM              56.6       45.3       28.1    47.3
>  cRT                 61.7       45.9       26.8    49.4
>  tau-norm      59.1       46.9       30.7    49.4
> -------------------------------------------------------
>  cos                 60.4       46.8       29.3    49.7
>  cos(noRelu)  60.7       46.9      28.0    49.6
> -------------------------------------------------------
> The above discussion has been included in Appendix B.7 of our revised manuscript.

---

### Official Review · AnonReviewer2 · 2019-10-22
**Official Blind Review #2**

**Rating:** 8

**Review:**

The paper tries to handle the class imbalance problem by decoupling the learning process into representation learning and classification, in contrast to the current methods that jointly learn both of them. They comprehensively study several sampling methods for representation learning and different strategies for classification. They find that instance-balanced sampling gives the best representation, and simply adjusting the classifier will equip the model with long-tailed recognition ability. They achieve start of art on long-tailed data (ImageNet-LT, Places-LT and iNaturalist).

In general, this is paper is an interesting paper. The author propose that instance-balanced sampling already learns the best and most generalizable representations, which is out of common expectation. They perform extensive experiment to illustrate their points.

--Writing:
This paper is well written in English and is well structured. And there are two typos. One is in the second row of page 3, "… a more continuous decrease [in in] class labels …" and the other one is in the first paragraph of section 5.4, "… report state-of-art results [on on] three common long-tailed benchmarks …".

--Introduction and review:
The authors do a comprehensive literature review, listing the main directions for solving the long-tailed recognition problem. They emphasis that these methods all jointly learn the representations and classifiers, which "make it unclear how the long-tailed recognition ability is achieved-is it from learning a better representation or by handling the data imbalance better via shifting classifier decision boundaries". This motivate them to decouple representation learning and classification.

--Experiment:
Since this paper decouples the representation learning and classification to "make it clear" whether the long-tailed recognition ability is achieved from better representation or more balanced classifier, I recommend that authors show us some visualization of the feature map besides number on performance. Because I am confused and difficult to image what "better representation" actually looks like.

The authors conduct experiment with ResNeXt-{10,50,101,151}, and mainly use ResNeXt-50 for analysis. Will other networks get similar results as that of ResNeXt-50 shown in Figure 1?

When showing the results, like Figure 1, 2 and Table 2, it would be better to mention the parameters chosen for \tau-normalization and other methods.

Conclusion:
I tend to accept this paper since it is interesting and renews our understanding of the long-tailed recognition ability of neural network and sampling strategies. What's more, he experiment is comprehensive and rigorous.

**Experience Assessment:**

I do not know much about this area.

**Review Assessment: Checking Correctness Of Derivations And Theory:**

N/A

**Review Assessment: Checking Correctness Of Experiments:**

I carefully checked the experiments.

**Review Assessment: Thoroughness In Paper Reading:**

I read the paper at least twice and used my best judgement in assessing the paper.

---

> ### Author Response · Authors · 2019-11-15
> **Response to Blind Review #2**
>
> A1: [Better representation visualization]
> In general, “better representation” in computer vision means the learned features are discriminative and can be useful to improve a subsequent task, e.g. image classification. Under this context, the cRT and tau-normalization classification results with different representation learning strategies are strong evidence that representation learned with instance balanced sampling are the best.
>
> A very common way to evaluate the quality of representation is to apply a non-parametric classifier (usually a KNN) on top of the representation. It is worth noting that our NCM classifier essentially is a *non-parametric nearest neighbor classifier*. The consistent results observed with NCM provide a direct support of our argument.
>
> We agree that providing some intuitive interpretation of the learned representations would be  helpful. We tried visualizing the feature maps but found it not informative. We take this as an open research problem and welcome new suggestions.
>
> A2: [Results with different backbones] Will other networks get similar results as that of ResNeXt-50 shown in Figure 1?
>
> Yes, the same conclusion holds with different backbones, please refer to Table.7 in the paper for details.
>
> A3: [tau normalization] Please refer to general response.

---

### Official Review · AnonReviewer1 · 2019-10-24
**Official Blind Review #1**

**Rating:** 6

**Review:**

The paper considers the problem of long-tailed image classification, where the class frequencies during (supervised) training of an image classifier are heavily skewed, so that the classifier underfits on under-represented classes. Different known and novel sampling schemes during training as well as post-training procedures to restore the class balance after training are studied. The overall best strategy turns out to be naive training on the skewed training set, and post-hoc rebalancing only of the classification stage. The paper presents various ablation studies and comparisons with related methods on the ImageNet-LT, Places-LT, and iNaturalist data sets, achieving state-of-the-art performance.

The paper is well-written and gives a nice overview on related work and in particular reweighted sampling schemes. The proposed methods and variations appear to be simple, yet very effective, and the insight that decoupling representation and classifier learning performs well on long-tailed classification seems novel. The experiments are mostly thorough and detailed. Here are some more detailed comments:

- My main concern is the selection strategy of \tau in \tau-normalized classification. The authors merely specify that they choose it in the interval (0,1). How is this tuned exactly? Per data set or the same for all data sets? On a validation set? It would be great to provide more details and guidelines for practitioners. Also, in Fig. 2 left, what is the \tau used?

- It would be interesting to see whether the performance can be improved by training a shallow MLP rather than just retraining the weights of the linear classifier W.

- Retraining a linear classifier on a fixed representation can be brittle, at least this can be observed in the context of unsupervised representation learning. The authors should add details about the exact schedules, batch size etc. used for retraining the linear classifier in cRT.

Overall I like the paper, but it is important to add more detail, in particular about the choice of of \tau.


**Experience Assessment:**

I have read many papers in this area.

**Review Assessment: Checking Correctness Of Derivations And Theory:**

I assessed the sensibility of the derivations and theory.

**Review Assessment: Checking Correctness Of Experiments:**

I carefully checked the experiments.

**Review Assessment: Thoroughness In Paper Reading:**

I read the paper at least twice and used my best judgement in assessing the paper.

---

> ### Author Response · Authors · 2019-11-15
> **Response to Blind Review #1**
>
> A1: [tau normalization] Please refer to the general response
>
> A2: [MLP v.s. Linear Classifier]
> We experimented with MLPs with different layers (2 or 3) and different number of hidden neurons (2048 or 512). We use ReLU as activation function, set the batch size to be 512, and train the MLP using balanced sampling on fixed representation for 10 epochs with a cosine learning rate schedule, which gradually decrease the learning rate to zero. We conducted experiments on two datasets.
>
> On ImageNet-LT, we use ResNeXt50 as the backbone network. The results are summarized in the following table. We can see that when the MLP going deeper, the performance are getting worse. It probably means the backbone network is enough to learn discriminative representation.
> -----------------------------------------------------------------------------------------
>                          hid_dim=2048             |              hid_dim=512
> Layers  ------------------------------------------------------------------------------
>              Many  Medium   Few      All  |  Many  Medium   Few     All
> -----------------------------------------------------------------------------------------
>      1       61.7      45.9        26.8    49.4
>      2       60.8      44.4        24.5    48.0      59.9      44.3        25.1   47.7
>      3       60.3      44.3        23.7    47.7      59.3      43.7        23.9   47.0
> -----------------------------------------------------------------------------------------
>
> For iNaturalist, we use the representation from a ResNet50 trained for 200 epochs. We only consider a hidden dimension of 2048, as this dataset contains much more classes. The results show that performance drop is even more severe when a deeper classifier is used.
> -------------------------------------------------
> Layers  Many  Medium   Few     All
> -------------------------------------------------
>      1       73.2     68.8      66.1    68.2
>      2       60.4     61.8      60.6    61.2
>      3       68.5     63.6      60.1    62.8
> -------------------------------------------------
> Above discussion can be found in Appendix B.6 of our revised manuscript.
>
> A3: [Details on retraining (cRT)]
> For classifier retraining, we use random initialization, cosine learning rate schedule (decrease from 0.2 to 0), the batch size is 512, and balanced sampling.
> We will add missing experimental details to the paper. As we stated in the general response, we also promise to release all the code, model and training recipes to facilitate reproducible research in long-tailed recognition.

---

### Author Response · Authors · 2019-11-15
**Response To All Reviewers: On Tau-Normalization**

We thank all reviewers for their valuable feedback; we are glad to see that all reviewers agree that our findings for long-tailed recognition are surprising and insightful. We also note that we intend to release all the code, models and training recipes to facilitate reproducible research in long-tailed recognition.

We noticed that most of the concerns come from tau-normalization. We thank the reviewers for making us look deeper into this strategy; we’d like to clarify the setup and discuss some new findings inspired by the high-quality reviews here.

First we would like to clarify *how tau is selected*. In our submission, tau is determined by grid search on a validation dataset. The search grid is [0.0, 0.1, 0.2, ..., 1.0]. We use overall top-1 accuracy to find the best tau on validation set and use that value for test set.

The best tau value should reflect the category imbalance degree of a dataset, thus it is expected that it should not be identical across different datasets. In our experiments, we use tau value 0.7 on ImageNet_LT, 0.8 on Places_LT and 0.3 on iNaturalist. Though tau is dataset-dependent, we find it is very robust to backbone networks, training epochs and other optimization hyper-parameters: we verified that best tau values are identical among those different settings.

The current tau-normalization strategy does require a validation set to choose tau, which could be a disadvantage depending on the practical scenario. Can we do better?

== Finding tau value on training set==
After reading the reviews, we made an attempt to select tau directly on the *training* dataset. Surprisingly, final performance on testing set is very similar, with tau selected using training set only.

We achieve this goal by simulating a balanced testing distribution from the training set. We first feed the whole training set through the network to get the top-1 accuracy for each of the classes. Then, we average the class-specific accuracies and use the averaged accuracy as the metric to determine the tau value. In the following table, we compare the tau found on training set and validation set for all three datasets. We can see that both the vale of tau and the overall performances are very close to each other, which demonstrates the effectiveness of searching for tau on training set. This strategy offers a practical way to find tau even when validation set is not available.

---------------------------------------------------------------------------------------
Dataset             split    learn   tau    |   Many  Medium   Few   All
---------------------------------------------------------------------------------------
ImageNet_LT   val         N      0.7            59.1     46.9      30.7    49.4
                           train     N      0.7             59.1     46.9      30.7    49.4
                           train     Y       0.6968      59.2     46.9      30.6    49.4
---------------------------------------------------------------------------------------
iNaturalist        val         N      0.3            65.6     65.3      65.9    65.6
                           train      N      0.2            69.0     65.2      63.6    65.0
                           train      Y       0.3146     65.1     65.2      66.1    65.6
---------------------------------------------------------------------------------------
Places_LT          val        N       0.8            37.8     40.7      31.8    37.9
                           train     N      0.6             41.4     39.3     25.3    37.4
                           train     Y       0.5246      42.6     38.3     22.7    36.8
---------------------------------------------------------------------------------------

== Learning tau value on training set==
We further investigate if we can automatically learn the tau value instead of grid search. To this end, following cRT, we set tau as a learnable parameter and learn it on the training set with balanced sampling,  while keeping all the other parameters fixed (including both backbone network and classifier). Also, we compare the learned tau value and the corresponding results in the above table (denoted by ‘learn’ = ‘Y’).
This further reduces the manual effort of searching best tau values and make the strategy more accessible for practical usage. We will incorporate these new findings in the paper, and once again, we thank all reviewers for the inspiring comments. All above discussion is updated to our manuscript in Appendix B.5.

---

### Public Comment · ~Vincent_Tao_Hu1 · 2020-04-12
**Where is B.5**

Hi, thank for your great work, where is B.5? I cannot find it in your current version.

---

### Public Comment · ~Not_Use_Anymore1 · 2020-05-11
**Question about the results**

Hi, I think this is an inspiring work. I just want to ask a question about the results.

Correct me if I'm wrong: in Figure 1, class-balanced joint training means class-balanced representation learning and class-balanced classifier re-training, which should have the same results. That is to say, in Figure 1, line "Class-balanced", "Joint" and "cRT" results should be the same.

However, it is not the case in Figure 1.

Do I misunderstand it? Or it is because of the randomness of experiments.

---

> ### Author Response · Authors · 2020-05-12
> **"Joint" and "cRT" are two different classifiers**
>
> As you might know, in our paper, there are two different training stages, i.e., representation learning and classifier learning.
>
> In representation learning, softmax cross-entropy loss is applied after the last FC layer, which is called the "Joint" classifier in our paper. In the classifier retraining, the backbone network is fixed while the last FC layer is processed in multiple ways, either retrained or tau-normalized.
>
> Therefore, for "Class-balanced sampling", "Joint" means the classifier learned during representation learning, while "cRT" means the classifier retrained during classifier retraining. So it's possible that the results are slightly different.

---

### Public Comment · ~Guangxiang_Zhao3 · 2022-04-25
**Re-weighting at the sample level (CE loss down-weights the gradients of well-classified samples）is also harmful to the representation learning, not just by reweighting at the class level.**

Hi,

Thanks for your insightful work. Your work finds that re-weighting at the class level ( down-weight the learning of head classes) according to the frequency table harms representation learning in the imbalanced scenario.

In my recent work [1], I find that CE Loss down-weights the learning of well-classified samples by making their gradients smaller (the differential of CE Loss: -1logp is -1/p), and this common practice is also harmful to the representation learning in imbalanced and balanced scenarios. Moreover, down-weighting the learning of well-classified samples particularly harms the accuracy of rare classes.

Therefore, in the stage of representation learning, we should not reweight at the class level (substituting reweighting/resampling with CE Loss as your paper did), and not to reweighting at the sample level (substitute CE Loss with a new loss that does not down-weight the learning of well-classified examples).


[1]  Well-classified Examples are Underestimated in Classification with Deep Neural Networks, AAAI 2022

---

### Decision · Program_Chairs · 2019-12-19

**Decision:**

Accept (Poster)

**Comment:**

This paper presents an approach for the long-tailed image classification, where the class frequencies during (supervised) training of an image classifier are heavily skewed, so that the classifier underfits on under-represented classes. The authors' responses to the reviews clarified most of their  concerns, although some reviewers pointed out that some of the details regarding experiments such as the construction of the validation set and the selection of balanced/imbalanced sets remain unclear. Overall, we believe this paper contains interesting observations to be shared.